# An Approach to the Use of Glycol Alkoxysilane–Polysaccharide Hybrids in the Conservation of Historical Building Stones

**DOI:** 10.3390/molecules26040938

**Published:** 2021-02-10

**Authors:** Miguel Meléndez-Zamudio, Ileana Bravo-Flores, Eulalia Ramírez-Oliva, Antonio Guerra-Contreras, Gilberto Álvarez-Guzmán, Ramón Zárraga-Nuñez, Antonio Villegas, Merced Martínez-Rosales, Jorge Cervantes

**Affiliations:** División de Ciencias Naturales y Exactas, Departamento de Química, Universidad de Guanajuato, Guanajuato 36000, Mexico; melendem@mcmaster.ca (M.M.-Z.); ie.bravoflores@ugto.mx (I.B.-F.); eraoliva@ugto.mx (E.R.-O.); ja.guerra@ugto.mx (A.G.-C.); g.alvarez@ugto.mx (G.Á.-G.); rzarraga@ugto.mx (R.Z.-N.); vigaja@ugto.mx (A.V.); mercedj@ugto.mx (M.M.-R.)

**Keywords:** THEOS, MeTHEOS, chitosan, stone conservation, siliceous and calcareous stones

## Abstract

Stone consolidants have been widely used to protect historical monuments. Consolidants and hydrophobic formulations based on the use of tetraethoxysilane (TEOS) and alkylalkoxysilanes as precursors have been widely applied, despite their lack of solubility in water and requirement to be applied in organic media. In the search for a “greener” alternative based on silicon that has potential use in this field, the use of tetrakis(2-hydroxyethyl)silane (THEOS) and tris(2-hydroxyethyl)methyl silane (MeTHEOS) as precursors, due their high water solubility and stability, is proposed in this paper. It is already known that THEOS and MeTHEOS possess remarkable compatibility with different natural polysaccharides. The investigated approach uses the water-soluble silanes THEOS–chitosan and MeTHEOS–chitosan as a basis for obtaining hybrid consolidants and hydrophobic formulations for the conservation of siliceous and calcareous stones. In the case of calcareous systems, their incompatibility with alkoxysilanes is known and is expected to be solved by the developed hybrid consolidant. Their application in the conservation of building stones from historical and archeological sites from Guanajuato, México was studied. The evaluation of the consolidant and hydrophobic formulation treatment was mainly conducted by determining the mechanical properties and contact angle measurements with satisfactory results in terms of the performance and compatibility with the studied stones.

## 1. Introduction

The conservation of stones in historical buildings around the world is receiving a growing amount of interest due to the importance of preserving historical memory for new generations. Different authors have highlighted the importance of stone conservation using varied scientific approaches due to the complexity and diversity of problems that require solutions. Stone conservation is an area where chemistry, physics, material science, biology, history, architecture, archeology, restoration, geology, etc. find a common point of confluence [1,2,3,4].

In an overview of research on stone conservation, the process referred to as consolidation is considered to be an active conservation process “where stone is severely weakened by decay, and some form of consolidation may be necessary to restore some strength. Ideally, one might hope to make the stone at least as strong as it was originally, so it might resist further decay” [5]. The same overview points out that “some of the causes of stone decay are sudden and rapid in their effect. Those toward the latter part of the following list are slow and more insidious: earthquake, fire, flood, terrorism, vandalism, neglect, tourism, previous treatments, wind, rain, frost, temperature fluctuations, chemical attack, salt growth, pollution, biodeterioration, intrinsic factors, and so on” [5].

Stone consolidants have been widely used to protect historical monuments. Commercial products containing alkoxysilanes, such as tetraethoxysilane (TEOS), are commonly applied as consolidants for decayed natural stones. They are applied as low viscosity monomers or dimers in solutions that may include water, ethanol, or other organic solvents (generally MEK and acetone). An organometallic catalyst such as di-n-butyltin dilaurate (DBTL) is commonly used to increase the rate of polycondensation. Hypothetically, the fluid deeply infiltrates the porous network of the stone and, through a sol–gel process, forms a silica gel that works as a new cement for the matrix. This is done to re-establish the cohesion between loosened material grains as well as to restore the original mechanical resistance in the decayed material. The use of alkoxysilane-based products as stone consolidants to conserve decayed quartz-bearing rocks, such as sandstone, or siliceous natural materials in general, has been a common practice for several decades [6], although the first patent was approved much earlier in 1924 [7].

However, in the case of calcareous stones, their non-compatibility with alkoxysilanes has been established, so several approaches have been investigated [8,9,10].

In a recent review regarding the application of alkoxysilanes in the field of stone conservation, it is pointed out that conventional alkoxysilane-based consolidants have several drawbacks that hinder their successful application in carbonate stones. In terms of addressing these issues, the modification of alkoxysilanes has resulted in some improvements. The cracking tendency of alkoxysilanes has been solved by the introduction of elastic segments, surfactant templates, and nanoparticle loading, in addition to other elements. Nevertheless, there is still much room for improvement. The complexity of sol–gel chemistry and the conceptual incompatibility between alkoxysilane-based consolidants and carbonate minerals do not allow for successful prediction of the consolidation behavior. The proposed consolidants often render the treated stone hydrophobic, which is interesting if the objective is to obtain both a hydrophobic and consolidation treatment, although the risk of incompatibility exists. Despite the drawbacks, the study of new alkoxysilane-based consolidants that can provide multifunctional (consolidant, self-cleaning, or biocide) properties is still being conducted. The development of such properties again makes alkoxysilanes a feasible alternative for consolidating carbonate stones in important stone heritage if their compatibility and durability are effective. Of course, any consolidation action should not hinder or limit future interventions. In addition, due to the complexity of the interaction between the alkoxysilanes and the stone, a more detailed study of the mechanism is needed [11].

A recent investigation addresses the impact and importance of the carbonate medium in the sol–gel processes of stone consolidation based on alkoxysilanes and the possible detrimental effects in practical applications, highlighting the need for the design and development of new alkoxysilane-based consolidants to consider this effect [12].

The interest of our research group in the use of alkoxysilanes in the conservation of historical building stone has led to investigations on many topics, such as chemical, physical, geological, and mineralogical characterization of historical building stones; considering decay and biodecay evaluation; and the synthesis and application of consolidant and hydrophobic formulations based on alkoxysilanes in siliceous stones, with the aim of solving reported problems in the performance of some of the commercial formulations through the development of hybrid systems based on TEOS/SiO_2_St/PDMS-OH. Most of our work in the field has focused on siliceous stones [13,14,15,16,17,18,19,20,21].

Tetraethoxysilane (TEOS) and alkylalkoxysilanes are widely used as precursors in consolidant and hydrophobic formulations. Due to their lack of solubility in water, the formulation must be applied in organic solvents (VOCS), promoting a clear disadvantage in terms of “green chemistry” in this field. In the search for “greener” silicon derivatives with potential use as consolidants, the use of tetrakis(2-hydroxyethyl)silane (THEOS) and tris(2-hydroxyethyl)methyl silane (MeTHEOS) as precursors is suggested because of their remarkable water solubility. Initial studies of the THEOS precursor were conducted by Mehrotra and Narain [22] and, subsequently, introduced by Hoffmann and his group through the transesterification reaction of TEOS and ethylene glycol. The complete characterization of properties achieved by Hoffman demonstrated that THEOS possesses high solubility and stability in water and, as a result, the use of typical organic solvents is suppressed [23].

On the other hand, in 2004, Shchipunov, Tatyana, and Karpenko reported the compatibility of THEOS with different natural polysaccharides, including chitosan. As observed by the authors, polysaccharides worked as accelerators, catalysts, and templates for the silica generated in situ by the sol–gel process; modification of the synthesis conditions led to different properties and structures suitable for obtaining monolithic hybrid materials. For the THEOS–chitosan system, a transparent hybrid material was obtained, and no phase separation or syneresis was detected [24,25,26,27].

An important aspect of the hybrid THEOS–chitosan was described when it was shown that it accelerated, catalyzed, and served as a template for silica generated in situ by sol–gel, thus manipulating its synthesis as well as the properties and structure of the produced monolithic hybrid materials [24]. The potential application in drug encapsulation is an example of a very important and actual field of investigation where these biocompatible hybrid materials can have an impact [28]. The hydrolysis of THEOS produces silicic acid. As a result of the complete compatibility with chitosan, the presence of sol–gel in solution results in gelation of the non-gellable chitosan as well as other polysaccharides. For example, the gelation by mineralization of carbohydrate macromolecules strengthened them and provided cross-linking [25]. The gelation time required for the sol–gel transition and the dynamic rheological properties of the resultant gel matrix could be modulated by the amount of added THEOS. The hybrid material has found application in electrochemical biosensors [27,29,30] and is able to obtain stable and intact thin films or monolithic hybrid gels [31].

On the other hand, it is important to consider that the hybrid has the presence of primary amines and hydroxyl groups in the chitosan structure, and such functionalities have a key effect on the biopolymer solubility. Additionally, they can act as reactive sites for the covalent interaction via condensation in the chitosan–siloxane network. In this direction, we have recently reported a detailed study to elucidate the covalent interaction between reactive silanols (from the complete hydrolysis of THEOS and MeTHEOS) and chitosan. The results suggest that the condensation site forming the silyl–ether bond is located at C6 of the chitosan structure [32].

The characteristics developed by the hybrid system THEOS–chitosan enable us to suggest a new application in the area of conservation of historical stone buildings that is based on the preparation of hybrid consolidant and hydrophobic formulations that can be applied in the conservation of stones with a siliceous and calcareous composition; THEOS–chitosan can be used in the consolidant formulation and MeTHEOS in the hydrophobic formulation. Bearing in mind that a great number of important historical monuments in different parts of the world have siliceous or calcareous stones as building materials, some advantages can be considered in the introduction of glycol alkoxysilane–chitosan in the field of stone conservation, i.e., that the “green” formulations based on water-soluble silanes do not need organic solvents for their application, these hybrid formulations can potentially be used as an alternative to solve the problem of compatibility between alkoxysilanes and calcareous stone, and the synergy originating from the alkoxysilane–chitosan interaction in terms of the film formation capacity of chitosan, as well as the antimicrobial activity, suggests the possibility to avoid the biodegradation of stones by many organisms. It is important to consider that the material aptitude to biological colonization by certain organisms is called the bioreceptivity and is dependent on different environmental factors, such as the pH, water availability, climate exposure, mineral composition, porosity, permeability, and nutrient sources [5,33].

The antimicrobial activity of chitosan has been widely studied; for example, the antibiofilm properties of chitosan-coated surfaces, where chitosan offers a flexible, biocompatible platform for designing coatings to protect surfaces from infection [34]. The state of the art of antimicrobial chitosan and chitosan derivatives and the effects of structural modifications on the activity and toxicity have been reviewed toward improving the understanding of the bioactivity and to develop more useful chitosan conjugates [35]. On the other hand, it has been pointed out that chitosan and its derivatives can be called environmental purification functional materials as they can effectively control the growth and reproduction of hazardous bacteria and also control toxic pollutants [36]. The antibacterial activity of chitosan extracted from a pen shell against both Gram-positive and Gram-negative bacteria was recently reported [37]. Furthermore, chitosan and its derivatives have been studied due to their antimicrobial properties in the context of preventing and treating denture stomatitis, which can be caused by fungi [38].

## 2. Results and Discussion

The results and discussion section considers two main aspects: synthesis and characterization of the formulations THEOS–chitosan and MeTHEOS–chitosan and their application in siliceous and calcareous building materials. Films of the silane–chitosan hybrids were obtained and characterized using different methods. Such characterization was useful for revealing the film behavior inside or on the stones as a result of the consolidation or hydrophobic treatment.

The precursors of the hybrid formulations—THEOS and MeTHEOS—were obtained according to the most reported and used method (the transesterification reaction of TEOS or MeTEOS with ethylene glycol) [22,23]. At the reaction conditions used (140 °C and 15 h of reaction time), high yields are obtained (83% to 92%). Nevertheless, the morphology of the product obtained from different syntheses under the same conditions is diverse (liquid, translucid viscous, or gel), without altering the water solubility. The analysis of the transesterification reaction products (THEOS and MeTHEOS) was conducted by ^29^Si-NMR in solution (DMSO-d6), in respect of the TEOS and MeTEOS spectra as the reference (singlet at −82.5 ppm and −43 ppm, respectively). As demonstrated in previous studies and by recent observations, THEOS does not exist as a single molecule, as can be seen from the diverse morphology that the isolated product exhibits, and various silicon species appear in the Q unit region (−81.7, −82.3, −83.6, and −88.6 ppm) for THEOS; for MeTHEOS (−41.7, −44.1, −49.8, −52.3, and −58.2 ppm), different peaks appear in the T unit region [32,39,40].

### 2.1. Synthesis and Characterization of Silane–Chitosan Hybrid Films

A very wide range of tests of THEOS–chitosan and MeTHEOS–chitosan solutions were prepared, using different proportions of the reagents, in order to find out the most appropriate formulations to be applied in stone treatment. The selection of formulations to be used was carried out through observation of the film characteristics obtained in terms of the flexibility, homogeneity, transparency, and resistance to syneresis, where the excellent capability of chitosan as a film formulation was a key aspect in terms of the concentration of chitosan used. No phase separation was observed. The extensive testing revealed that a selection of formulations with similar physical characteristics were obtained, offering the possibility to apply them in experiments with different goals (see Materials and Methods). For example, the selected formulation for the films characterized by Fourier-Transform Infrared Spectroscopy-Attenuated Total Reflectance (FTIR-ATR), Scanning Electron Microscopy-Energy Dispersed X-ray Spectroscopy (SEM)–EDX, thermal stability, and solid-state Nuclear Magnetic Resonance (NMR) analysis was based on 10 mL of an aqueous solution of chitosan (0.5% in acetic acid at 1% with 72% deacetylation) and 0.5 g of THEOS. Some variations in the formulation composition were used in several analyses, such as hardness and contact angle determinations, in order to study the effect of the silane–chitosan ratio (formulations referred to as 1, 2, and 3 in Materials and Methods).

#### 2.1.1. FTIR

The FTIR spectra of THEOS–chitosan and MeTHEOS–chitosan films present similar characteristics in terms of their band frequencies for –OH (3500–3200 cm^−1^), –CH, –CH_2_, and –CH_3_ (3000–2840 and 1460–1350 cm^−1^), amide N–C=O (1655 cm^−1^ and 1580 cm^−1^), amine –NH_2_ (1320 cm^−1^), and the Si–O–Si network (1110–1000 cm^−1^). The main difference is due the presence of the bands at 1270 and 770 cm^−1^ (Si–CH_3_ and –C–Si–O, respectively) in the MeTHEOS–chitosan film (Appendix A).

#### 2.1.2. SEM–EDX

SEM–EDX of THEOS–chitosan and MeTHEOS–chitosan films are presented in Figure 1. The 10,000× amplification illustrates the films’ characteristics, showing that they are flexible, thin, and transparent and have no evident imperfections. The elements observed according to EDX analysis are carbon, nitrogen, oxygen, and silicon, in accordance with the hybrid composition.

#### 2.1.3. Thermal Stability of the Hybrids

The thermal stability of the hybrid films was studied. The films were exposed to different temperatures, from room temperature to 700 °C, and FTIR–ATR spectra were collected to determine any structural changes. Comparative spectra obtained at 25 °C and 350 °C are presented (Figure 2). As can be observed, the films are thermally stable until 350 °C. In the case of MeTHEOS–chitosan, the fragment –SiCH_3_ is removed around 500 °C.

#### 2.1.4. Structural Characterization by Solid State NMR

A more detailed structural characterization of the hybrid THEOS–chitosan and MeTHEOS–chitosan films was conducted by solid state ^13^C-NMR (CPMAS) and ^29^Si-NMR (MAS) and reported recently [32]. The structural analysis of chitosan by ^13^C-NMR was taken as a reference to point out that the C6, bonded to a terminal hydroxy group, is suggested to be the condensation site of THEOS and MeTHEOS, once the C6 is the most sterically favored for this purpose. The most evident change in the chemical environment corresponds to the region of the chemical shift of C6 (60.71 ppm), exhibiting 2.24 ppm of difference with respect to the C6 of the chitosan film (58.47 ppm). The ^29^Si MAS and CPMAS spectra of the THEOS and MeTHEOS–chitosan films were collected for complementary structural analysis (Appendix A).

### 2.2. THEOS–Chitosan and MeTHEOS–Chitosan Formulations Applied to Siliceous and Calcareous Historical Building Materials

In consideration of the expected compatibility with both siliceous and calcareous materials, the application of the hybrid silane–chitosan in the field of the conservation of historical building stones is suggested. A key aspect is the water base application, which forgoes the use of organic solvents. In order to obtain data regarding the performance of the consolidant and hydrophobic formulations, different characterization methods were used, including FTIR, SEM–EDX, hardness determination, water absorption, and measurement of the contact angle (dynamic and static) and surface free energy. Different formulations in terms of the concentration of silane–chitosan were applied in the three selected materials (caliche, Sostenes, and Compañía). The different percentages of chitosan deacetylation were considered a variable to take into consideration in different determinations (i.e., hardness and contact angle measurements).

The FTIR–ATR spectra of the three studied materials were obtained. Figure 3 illustrates the caliche without treatment (Figure 3a) and after the consolidation (Figure 3b) and hydrophobic treatment (Figure 3c). The IR spectrum shown is typical of a calcite (1550–1350 and 872 cm^−1^ corresponding to stretching and bending vibrations, respectively). The bands at 1052 cm^−1^ and 715 cm^−1^ can be assigned to Si–O stretching due to the presence of a small concentration of silicates. After the consolidation treatment, the most important modifications in the spectrum are an increase in the intensity and broadness of the band at 1096 (Si–O–Si) and 722 cm^−1^, associated with the siloxane network; in the case of the sample treated with the hydrophobic formulation, a new small band at 1270 cm^−1^ appears, corresponding to the Si–CH_3_ fragment.

In the FTIR–ATR spectra of the Compañía stone, in accordance with its mineralogical composition (see Materials and Methods), bands are displayed at 1000, 1096, and 790 cm^−1^, characteristic of cristobalite, while those at 1000, 790, and 742 cm^−1^ correspond to feldspars and quartz. The most significant modifications after the consolidation and hydrophobic treatment, as in the previous caliche sample, occurred in the region associated with the Si–O–Si network, where the band is more intense and broader (1000 cm^−1^) and the small band at 1272 cm^−1^ (–SiCH_3_ fragment) appears. Due the siliceous composition, Sostenes and Compañía stones present similar spectra as a result of the application of the formulations (Appendix A).

Figure 4 (Appendix A), Figure 5 (Appendix A), and Figure 6 (Appendix A) present the results of SEM–EDX analysis of the stones before and after treatment. The order is caliche, Compañía, and then Sostenes.

In terms of the consolidation process, the aggregation of particles is evident because of the effect of the consolidant treatment. Nevertheless, the most important morphological changes are shown in caliche. In Compañía’s sample, which is the least compact stone according to SEM and the one with a higher percentage of water absorption, the consolidation effect is not so evident likely due to the low quantity of added consolidant. However, as is discussed later on, the increment in hardness indicated a positive consolidation effect. On the other hand, in the Sostenes stone, the morphological change is evident; regarding the hydrophobic treatment, a coating is observed in the three stones, with an important reduction in the porosity compared with the untreated materials, though leaving the stone with enough porosity to “breathe”, which is the final purpose of hydrophobic treatments in the stone conservation field.

Regarding EDX analysis, Sostenes and caliche stones display an increment in the carbon and nitrogen atomic percentage following treatment. A plausible interpretation is that the chitosan chains are exposed to the surface, not just in the case of the consolidant, but in the hydrophobic treatment (the methyl groups are surface oriented). Additionally, it is interesting to observe that the nitrogen atomic percentage is higher in consolidated Sostenes stone than in caliche, suggesting that the interaction between the consolidant and caliche possibly occurs via the free amine group. On the other hand, in Compañía stone, silicon is the element with a major atomic concentration on surface, probably suggesting, in accordance with SEM, that not enough consolidant formulation was added.

#### 2.2.1. Hardness Determination

The effectiveness of treatment in terms of the mechanical properties was determined by hardness measurements in stones consolidated using the THEOS–chitosan formulation and was performed by indentation with a Shore D durometer. Three variables that influence the hardness increase were considered: the applied formulation (as a function of the silane/chitosan ratio); the nature of the stone; and the percentage of deacetylation of the chitosan used in the formulation (%DDA). A statistical analysis was conducted to evaluate the effect of each variable (not included here). In a next step, the Shore D hardness values were transformed to the most common hardness scale, such as Vickers, Brinell, and finally Mohs, in order to compare the hardness data obtained with respect to reference values of well-studied materials based on the Mohs scale.

The formulations named 1, 2, and 3 (see Materials and Methods) were used in hardness determination. The hardness was measured at four points of the samples before and after treatment to characterize the hardness percentage increase. Interesting results were obtained for every formulation; however, the treatment that remarkably increased the mechanical properties of the stones was formulation 2, which contains chitosan with 66% DDA (Table 1).

Table 1 and Table 2 present illustrative data on the Shore D hardness determination and transformation, first to Vickers and Brinell scales, and then (Table 2) from the Brinell to Mohs scale. In any case, the hardness increase is evident, with some important variations, where the influence of the formulation (silane/chitosan ratio) seems to have a certain effect. However, it is also important to bear in mind the different stone compositions. In the case of caliche, the increase in hardness is quite similar, having a major effect on the siliceous materials.

The hardness values transformed to the Mohs scale and reported in Table 2 indicate a hardness increment of one unit in siliceous materials, and in the case of caliche (formulations 1 and 2), even 2 units. In general, the most important increase in hardness occurred in caliche. In terms of the Mohs scale, the hardness values from 5 to 7 obtained for the samples range between apatite to orthoclase and quartz. The hardness studies indicate that the samples treated with THEOS–chitosan displayed an important increase in the mechanical properties of the three materials.

#### 2.2.2. Water Absorption

Water absorption was tested using the Karsten tube technique, and measurements were taken before and after the application of the hydrophobic treatment (MeTHEOS–chitosan) on the stones (Table 3). The stone samples subjected to treatment present different mineral composition and water absorption values.

The penetration of water in the Compañía stone was quite high (51%) and was reduced to 7% with the hydrophobic treatment; such behavior makes sense due to its high pore diameter (macropores) in comparison with the other stones. Sostenes samples, that also possess a siliceous composition, had a water absorption value of 29% before treatment with a reduction to 10% as a result of the hydrophobic treatment. The calcareous stone (caliche) from the archeological site with an initial water absorption value of 46% exhibited remarkable reduction to 23%.

#### 2.2.3. Contact Angle Measurements

The evaluation of the hydrophobic formulation MeTHEOS–chitosan was studied by static and dynamic contact angle measurements using the same formulations 1, 2, and 3 and the % of DDA of 66. Because of the natural existence of defects on certain materials, as is the case of the stones studied in the current investigation, it has been suggested that a static water contact angle does not necessarily characterize the intrinsic water wettability [41]. Dynamic contact angle determination in the three stones is presented. The dynamic contact angle was obtained by the degree of hydrophobicity calculated by the hysteresis, representing the difference as θ_R_ (receding angle) − θ_A_ (advancing angle). The hysteresis values and the average of three measurements in different surface sections of the three stones are reported in Table 4. The dynamic angle measurements indicate that the surfaces of the three stones studied display water repellency.

The static contact angle was measured in three mediums (water, diiodomethane, and formamide) to take into consideration the different contributions of polar and non-polar mediums. The information obtained in the three mediums was useful for calculating the surface free energy or free energy of hydrophobicity by using the Owens and Van Oss (acid–base) methods [42,43].

The results are presented in Table 5 (1, 2, and 3 correspond to the formulation applied).

The value obtained for static contact angle showed that hydrophobic properties were achieved after the application of the MeTHEOS–chitosan formulation. In terms of the static contact angle in water, all values are over 90°, with caliche as an exception (formulation 1, 89.1°). Some authors consider that 90° demarcation can generally be applied to classify hydrophilic and hydrophobic behaviors; however, they consider contact angles closer to 90° to be relatively hydrophobic and lower contact angles to be relatively hydrophilic [41].

#### 2.2.4. Determination of the Surface Free Energy

In general terms, a sample with a low surface energy will cause poor wetting (a high contact angle). The reason for this is that the surface is not capable of forming strong bonds, so there is little energetic reward for the liquid to break bulk bonding in favor of interacting with the surface. On the contrary, a high surface energy will generally cause good wetting with a low contact angle. A surface will always try to minimize its energy. This can be done by adsorbing a material with a lower energy onto its surface [42,43]. The energy surface data are presented in Table 6.

The data interpretation reported in Table 6 is based on the criterium of a low surface energy value of 40 mN/m as a reference to consider a hydrophobic surface, although it is dependent on the model used; in the Owens model, the interval ranges from 49 to 3.16 mN/m, while in the Van Oss (acid–base), it ranges from 47 to 0 mN/m, so a lower value means a more hydrophobic surface [42,43]. According to the 40 mN/m reference value, or either the Owens or Van Oss model, caliche (formulation 1) is the only sample considered not to be hydrophobic. Some stones have a very low surface energy value, in agreement with the static contact angle obtained; Compañía is the most hydrophobic, followed by Sostenes and, finally, caliche. Moreover, the energy surface data indicate that formulation 2, in some way, is the most appropriate in terms of the silane/chitosan ratio (1 g of THEOS and 10 mL of a 0.5% aqueous solution of chitosan, and 66% DDA).

## 3. Materials and Methods

### 3.1. Chemicals

Tetraethoxysilane (98.5%), triethoxymethylsilane (98.5%), CDCl_3_, DMSO-d6, CH3COOD, D_2_O, and chitosan (72% deacetylation) were obtained from Sigma Aldrich. The ethylene glycol (JT. Backer, 99%) was distilled prior to its use in each reaction.

### 3.2. Synthesis of THEOS and Preparation of THEOS/MeTHEOS–Chitosan Formulations

The synthesis of THEOS and MeTHEOS by transesterification reactions was performed in a dry nitrogen atmosphere using Schlenk techniques. The reaction system used a three neck round bottom flask, a Vigreux column, a condenser, and a collector flask.

#### 3.2.1. Synthesis of THEOS by the Transesterification of TEOS

Into a three neck round bottom flask purged under N_2_ flow, 4.3 mL of freshly distilled ethylene glycol (4.76 g, 0.0768 moles) was added. After 30 min under magnetic stirring, 4.3 mL of TEOS (4 g, 0192 moles) was added drop by drop. The first drop of ethanol collected was considered the reaction initiation. The reaction temperature was 140 °C and was stable until the reaction ended (15 h). The reaction crude was concentrated under the vacuo line.

#### 3.2.2. Synthesis of THEOS–Chitosan and MeTHEOS–Chitosan Solutions

Formulation solutions were prepared by the addition of 0.5 g of THEOS or MeTHEOS to 10 mL of an aqueous solution of chitosan (0.5%) in acetic acid (1%) under magnetic stirring until complete dissolution. The % of DDA of chitosan was 72% (Sigma Aldrich Química, S.L., Toluca, Mexico). The described solutions were used to obtain hybrid films that were characterized by FTIR, SEM–EDX, and solid state NMR and regarding their thermal stability, and then used in consolidation and hydrophobic treatments of the stones. The formulations named 1, 2, and 3 were prepared using chitosan obtained from the extraction of shrimp exoskeleton with three different degrees of deacetylation (%DDA): 62%, 66%, and 70%. Formulation 1 (0.5 g of THEOS or MeTHEOS and 10 mL of 0.5% aqueous solution of chitosan), formulation 2 (1 g of THEOS or MeTHEOS and 10 mL of a 0.5% aqueous solution of chitosan), and formulation 3 (0.5 g of THEOS or MeTHEOS and 10 mL of a 1% aqueous solution of chitosan) were applied to the stones and used in the hardness and contact angle determinations.

#### 3.2.3. Synthesis of THEOS–Chitosan and MeTHEOS–Chitosan Films

The formulation solution obtained in each case was dispersed on plastic Petri dishes and left to dry at room temperature. Further characterization was performed by FTIR and solid state NMR (^29^Si and ^13^C).

### 3.3. Analytical Methods

#### 3.3.1. NMR

^29^Si, ^13^C, and ^1^H-NMR spectra in solution were collected using a Bruker AVANCE III 500 MHz spectrometer (Probe BBO-S2 5 mm). The internal references used were TMS (0 ppm) and hexamethyldisiloxane (chemical shifts at 6.54, 2.90, and 0.04 ppm in D_2_O, and 6.54, 1.96, and 0.06 ppm in DMSO-d6). In the case of ^29^Si, the one-dimensional sequence and inverse decoupling with a 90° pulse was used (d1 from 2 to 5 s, dt = 30 ms, ds = 4, and ns = 512). The ^13^C-NMR spectra were recorded using the one-dimensional sequence and proton decoupling with a 30° pulse (d1 = 1 to 2 s, dt = 30 ms, ds = 4, and ns = 128), and ^1^H spectra were obtained with the one-dimensional sequence with d1 = 1 s, ds = 2, and ns = 16.

The ^29^Si MAS and CPMAS and ^13^C CPMAS NMR spectra were collected using a Bruker AVANCE III 400 MHz spectrometer (probe: HRMAS 4 mm) using talc (−90 ppm as the reference for ^29^Si) and adamantane (28.46 and 37.52 ppm for CH and CH_2_, respectively). The parameters for the chitosan ^13^C CPMAS experiment were ns = 4096 and d1 = 4 s; for the THEOS/chitosan hybrid ^13^C CPMAS experiment, ns = 8192 and d1 = 4 s; for the THEOS/chitosan hybrid ^29^Si CPMAS experiment, ns = 8192 and d1 = 4 s; and for the THEOS/chitosan hybrid ^29^Si MAS experiment, ns = 14,336 and d1 = 6 s. The samples were placed in 4.0 mm zirconia rotors with a spinning rate of 8 kHz.

#### 3.3.2. FTIR Analysis

The spectra of hybrid films from 4000 cm^−1^ to 650 cm^−1^ were collected using a Perkin Elmer Spectrum FTIR 1600 coupled with an ATR accessory (germanium point, 100 μm in diameter). An average of 16 scans was obtained, with a resolution of 4 cm^−1^. Similar experimental conditions were used in the case of treated and untreated stone samples. The FTIR spectra were obtained from powders (−100 mesh) of each stone.

#### 3.3.3. Scanning Electron Microscopy (SEM)

A palladium–gold alloy was vacuum evaporated on the dried samples. The outer surfaces of the treated stones were then studied using a EVO15-HD ZEISS scanning electron microscope at a 15 kV accelerating voltage under various magnifications (1000, 5000, and 10,000).

#### 3.3.4. Stone Materials and Treatment

Samples of partially decayed siliceous stone (pink tuff) with the measurements 5 cm × 3 cm × 1 cm were collected from three different monuments or locations; two of them correspond to a tuff with a siliceous composition and are from two different historical monuments located in the city of Guanajuato, México (UNESCO World heritage City since 1988). The first is from the basement of a middle 20th century monument, called the statue of General Sostenes Rocha (who fought against the French army in México in the second half of the XIX century). Mineralogical analysis and XRD showed that the composition of the stone is mainly alkaline feldspars (46%), quartz (27%), mica (10%), kaolinite (9%), and smectite (3%), with traces of hematite. The second is from the church known as Oratorio de San Felipe Neri (traditionally called Compañía), which is a religious historical monument from the middle of the XVIII century [33]. The reported mineralogical analysis indicated alkaline feldspar (65%), quartz (29%), calcium silicate (3%), and hematite (4%) contents [44]. Additionally, a third sample from an archeological site called “Cerro de Los Remedios” (located in Comonfort county, Guanajuato state, México) was studied. As part of the basement of a pyramid, this stone possesses a calcareous nature (caliche), with CaCO_3_ (93%), CaO (1.5%), halloysite (5%), and traces of hematite. The relatively high amount of kaolinite present in the stone might be taken as evidence that some of the original feldspars have been hydrolyzed to clays by the weathering process known as kaolinization [45]. A comparative composition of the samples is presented in Table 7.

To evaluate the consolidation effect of the hybrid formulations, some samples (already cleaned and dried) were treated with the THEOS–chitosan formula, and to test the hydrophobic properties, others were treated with the MeTHEOS–chitosan formulation. The formulations were applied on stone samples by brushing in one phase under laboratory conditions until saturation.

Then, all samples were carefully wrapped in black plastic polypropylene sheets (as used in practical conservation) to permit gelling and aging for 2 weeks.

#### 3.3.5. Characterization of the Samples Treated with the Consolidant and the Hydrophobic Formulations

The hardness changes of the stones untreated and after treatment were measured as the Shore hardness using an REX 2000D indentation durometer. The hardness values obtained were transformed to Vickers, Brinell, and Mohs scales.

Water absorption measurements were carried out using the Karsten (Rilem) pipe [46]. The graduated pipe was fixed onto the sample and filled with water. Water absorption for each sample was measured as the difference between the quantities of water (mL) absorbed after five and thirty minutes.

Static and dynamic contact angles were measured using the OCA 15 Dataphysics system. Contact angle data obtained in the three classical liquids of different polarities (water, formamide, and diiodomethane) were used to calculate the surface free energy or free energy of hydrophobicity. The surface free energy was calculated from the OCA 15 Dataphysics software using the Owens and Van Oss (acid–base) methods [42,43].

## 4. Conclusions

As a result of the present investigation, a new approach to the application of glycol alkoxysilane–chitosan hybrids, including THEOS–chitosan and MeTHEOS–chitosan, in the area of stone conservation of historic stone buildings is presented and suggested, with THEOS–chitosan used as a consolidant and MeTHEOS–chitosan as a water repellent. Several aspects have been covered. First, a more detailed characterization of the hybrids has been described and discussed, as we believe that it was important to address the lack of such information. Once the hybrids had been obtained by a very simple synthetic procedure, they were applied to three different historic building stones and their performance was evaluated in detail. The application of the formulations to the stones is water-based, which implies the elimination of organic solvents, as an important contribution, but also suggestive of their use in the conservation of either siliceous or calcareous natural stones. By this means, synergetic benefits arise from the interaction of alkoxysilanes and chitosan in the performance of the formulations. The effectiveness of the consolidation and hydrophobic treatments was evaluated by different spectroscopic methods, such as FTIR–ATR and SEM–EDX, and physical analysis, such as hardness measurements, in the case of the consolidation, water absorption, characterization of the dynamic and static contact angle, and energy surface determination. The evaluation of the effectiveness of treatments is considered positive in terms of the consolidation and water repellency.

Several perspectives will now be presented for further study of the interaction of the formulations with the calcareous material, which in the present case, is caliche. A primary suggestion is that the interaction between the consolidant and caliche occurred via the free amino group of the chitosan. It is very important to point out that data on colorimetric changes after treatments have not been obtained, bearing in mind that the assessment of such analyses is quite important when stones of historical buildings are treated. No apparent colorimetric changes were observed after treatment in any sample, although this was recorded by simple observations. However, corresponding analyses must be performed and considered in perspectives; for example, the use of the Munsell method, which is commonly applied to observe chromatic variations as a result of sample treatment. Another interesting aspect that is currently under study, with preliminary results availables, is the use of the intrinsic fluorescence emission of chitosan. We have observed that such a property is maintained in the hybrid and might be a useful tool for ascertaining the effectiveness of the dispersion or penetration of the formulations on or inside the stone. Additionally, the antimicrobial property of chitosan is under study, which could lead to a formulation that will also prevent or solve cases of biodecay.

## Figures and Tables

**Figure 1 molecules-26-00938-f001:**
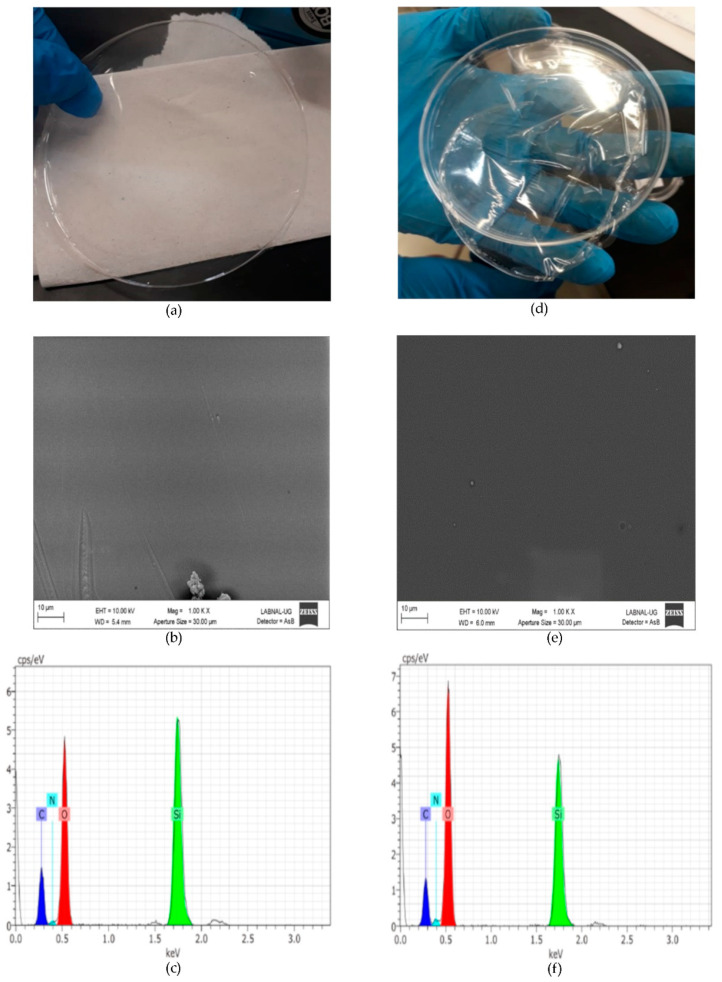
Characterization of the hybrid films. (**a**) MeTHEOS–chitosan film and Scanning electron microscopy (SEM) of the (**b**) tris(2-hydroxyethyl)methyl silane (MeTHEOS)–chitosan film; (**c**) EDX of the MeTHEOS–chitosan film; (**d**) THEOS–chitosan film and SEM of the THEOS–chitosan film (**e**); and (**f**) EDX of the THEOS–chitosan film.

**Figure 2 molecules-26-00938-f002:**
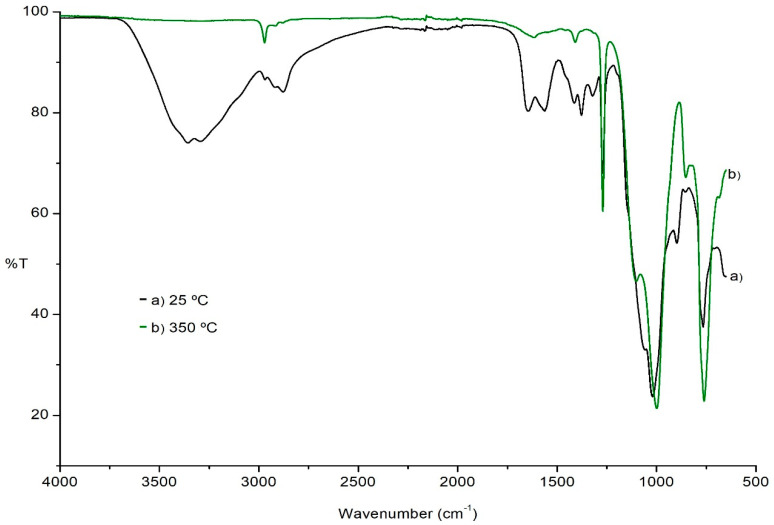
FTIR–ATR for a film of MeTHEOS–chitosan at (**a**) 25 °C and (**b**) 350 °C.

**Figure 3 molecules-26-00938-f003:**
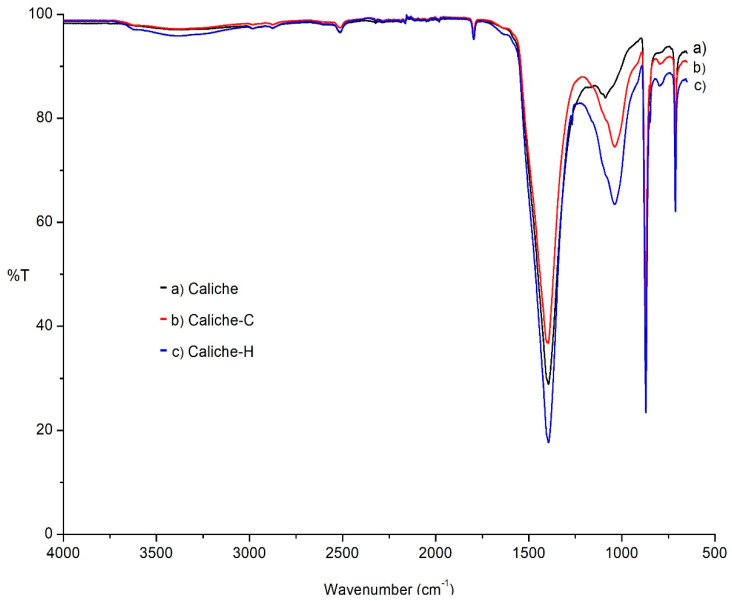
The FTIR–ATR spectrum of the calcite (**a**) without treatment, (**b**) consolidated, and (**c**) hydrofugated.

**Figure 4 molecules-26-00938-f004:**
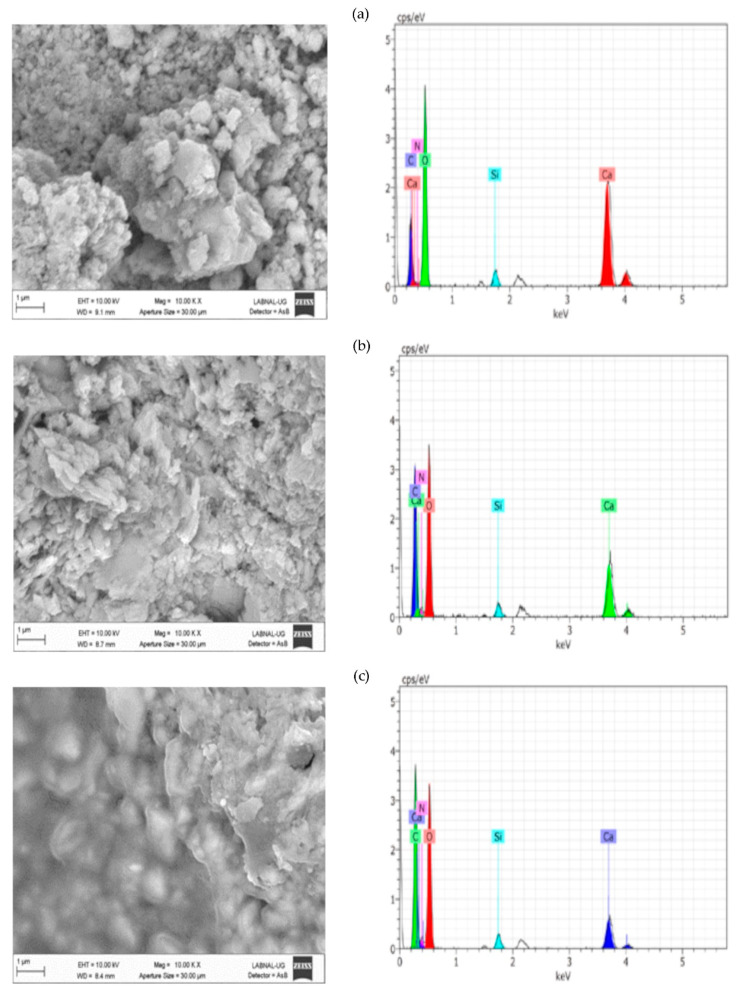
SEM–EDX analysis for the (**a**) caliche sample without treatment, (**b**) consolidated caliche sample, and (**c**) hydrophobic treated caliche sample.

**Figure 5 molecules-26-00938-f005:**
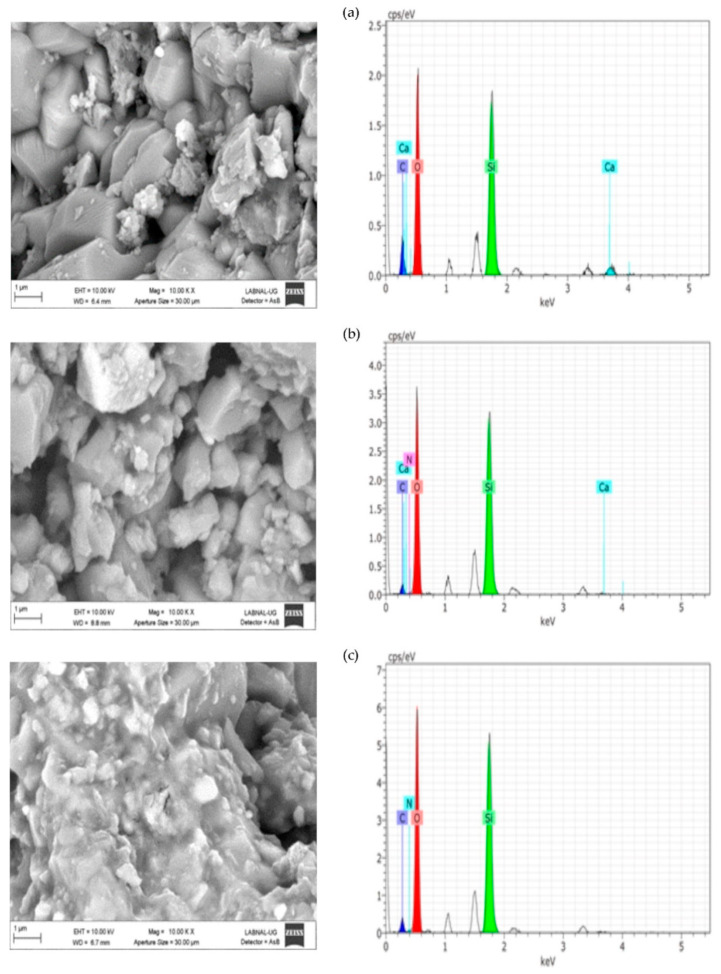
SEM–EDX analysis for the (**a**) Compañía sample without treatment, (**b**) consolidated Compañía sample, and (**c**) hydrophobic treated Compañía sample.

**Figure 6 molecules-26-00938-f006:**
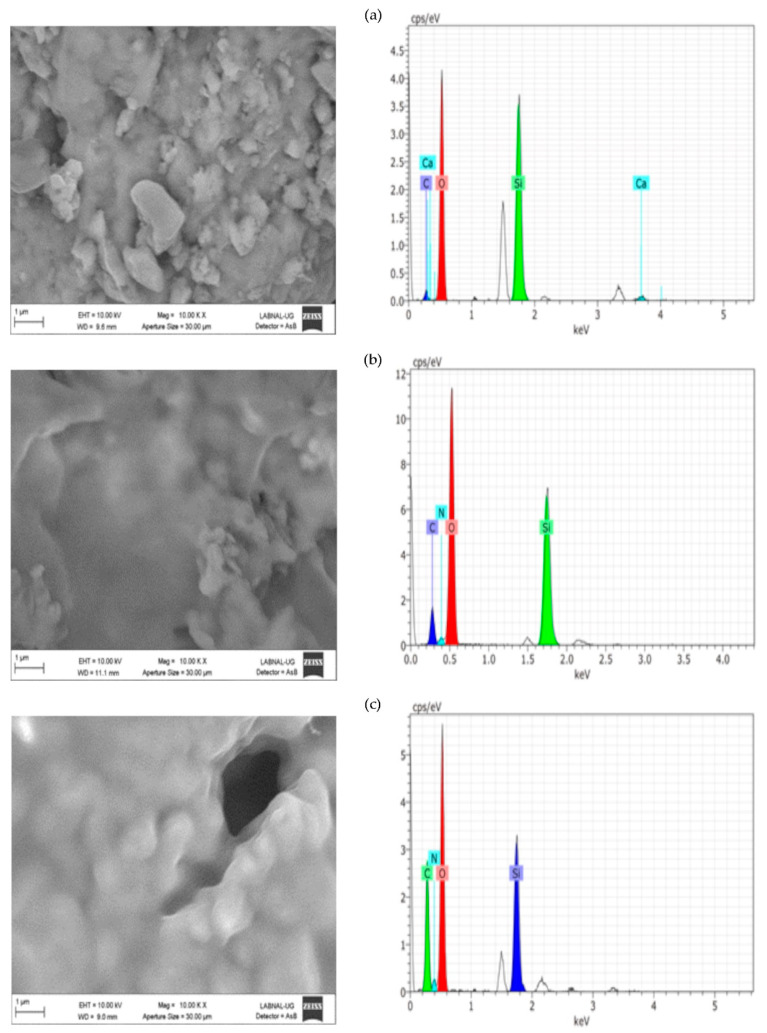
SEM–EDX analysis for the (**a**) Sostenes sample without treatment, (**b**) consolidated Sostenes sample, and (**c**) hydrophobic treated Sostenes sample.

**Table 1 molecules-26-00938-t001:** Shore D hardness and hardness increase (%) and the transformation to Vickers and Brinell scales (percentage of deacetylation of chitosan used in the formulation (%DDA) of 66).

%DDA	Stone	Formulation	Shore Hardness before	Vickers Hardness	Brinell Hardness	Shore Hardness after	Vickers Hardness	Brinell Hardness
66%	Caliche	1	71	581	511	81 (14%)	670	583
		2	70	564	497	81 (16%)	675	587
		3	72	585	515	81 (13%)	675	587
	Compañía	1	66	531	471	74 (12%)	604	530
		2	51	390	357	66 (29%)	531	471
		3	55	423	383	68 (23%)	550	486
	Sostenes	1	60	475	425	80 (33%)	663	577
		2	70	567	499	80 (14%)	665	579
		3	74	609	534	83 (12%)	687	597

**Table 2 molecules-26-00938-t002:** Hardness transformation from the Brinell to Mohs scale and % of hardness increase (66% DDA).

%DDA	Stone	Formulation	Brinell	Mohs	Brinell ^f,1^	Mohs ^f,1^
66%	Caliche	1	511	6	583 (14%)	8 (33%)
		2	497	6	587 (18.1%)	8 (33%)
		3	515	6	587 (13.9%)	7 (16.6%)
	Compañía	1	471	6	530 (12%)	7 (16.6%)
		2	357	5	471 (31.9%)	6 (20%)
		3	383	5	486 (26%)	6 (20%)
	Sostenes	1	425	6	577 (35.7%)	7 (16.6%)
		2	499	6	579 (16%)	7 (16.6%)
		3	534	6	597(11.7%)	7(16.6%)

^f,1^ = hardness measured after the consolidation treatment and increase percentage.

**Table 3 molecules-26-00938-t003:** Water absorption percentage on untreated and hydrophobic formulation (MeTHEOS)-treated stones.

Stone	Caliche	Caliche–MeTHEOS	Compañía	Compañía–MeTHEOS	Sostenes	Sostenes–MeTHEOS
Dry weight (g)	48.0	29.9	27.6	31.9	26.2	31.5
Wet weight (g)	70.1	33.1	41.7	34.2	33.9	34.7
Water absorption (%)	45.8	22.9	50.9	7.4	29.4	10.3

**Table 4 molecules-26-00938-t004:** Dynamic contact angle. Three contact angle measurements were performed at different points of the stones.

Stone		θ_A_ ^1^ (°)	θ_R_ ^2^ (°)	$\bar{\theta_{A}}\;(\text{°})$	$\bar{\theta_{R}}\;(\text{°})$	$\bar{\theta_{R}}-\bar{\theta_{A}}\;(\text{°})$
Caliche	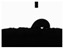	67.2	95.2	73.4 ± 5.5	109.5 ± 12.5	36.1
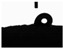	75.6	114.9
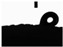	77.5	118.3
Compañía	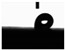	80.1	131.0	77.7 ± 1.2	113.6 ± 0.2	35.9
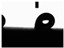	79.1	132.4
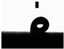	85.0	132.7
Sostenes	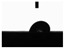	87.0	104.9	82.7 ± 3.7	107.5 ± 2.7	24.8
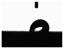	80.6	110.3
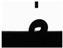	80.6	107.2

^1^ θ_A_ = advancing angle. ^2^ θ_R_ = receding angle.

**Table 5 molecules-26-00938-t005:** Static angle determinations for the treated stones.

Sample	Formulation	Medium		
		Water	Formamide	Diiodomethane
Caliche	1	89.1	83.3	54.2
	2	108.1	88.1	79.0
	3	96.2	92.1	70.6
Compañía	1	139.7	134.2	103.6
	2	135.4	132.1	109.3
	3	134.8	131.5	110.6
Sostenes	1	110.9	102.7	74.2
	2	116.3	89.0	84.8
	3	105.1	103.1	77.8

**Table 6 molecules-26-00938-t006:** Surface free energy of the stones treated with MeTHEOS–chitosan using the Owens and Van Oss (acid–base) methods ^1^.

Stone	Formulations	%DDA	Owens	Van Oss
Caliche	1	66%	46.9	48.6
	2		1.2	18.2
	3		33.6	19.5
Compañía	1		10.2	5.8
	2		8.9	4.2
	3		1.6	3.9
Sostenes	1		37.0	16.3
	2		0.00	17.1
	3		21.4	14.6

^1^ Surface energy units, mN/m.

**Table 7 molecules-26-00938-t007:** Mineralogical composition of the siliceous and calcareous stones.

Mineral	Caliche	Compañía	Sostenes
Alkaline feldspar	-	65%	46%
Quartz	-	29%	27%
Calcite	93%	-	-
CaO	1.5%	-	-
Mica	-	-	10%
Kaolinite	-	-	9%
Calcium silicate	-	3%	-
Halloysite	5%	-	-
Smectite	-	-	3%
Hematite	traces	4%	traces

## Data Availability

Not applicable.

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
