# Peer review of "An Approach to the Use of Glycol Alkoxysilane–Polysaccharide Hybrids in the Conservation of Historical Building Stones"

_molecules, 2021, doi:10.3390/molecules26040938_

Round 1

Reviewer 1 Report

The paper analyses the consolidant and hydrophobic formulations treatment using THEOS as greener product with respect the most commonly used Teos. The paper shows original and interesting data. The presentation of data is clear, and the general protocols are sound. Then I suggest the publication, even if some revisions are required.

At first, data on colorimetric changes after the treatments are not discussed and analyses are not done. The assessment of such potential is important, when an application to cultural heritage is welcomed. Therefore, two modification are required: The title: change into “Preliminary evaluation of Glycol......”and into the discussion of such further need should be discussed, including references.

The Author also at first introduce the potential of chitosan in reducing the biological colonization. Then they underline as further study will assess such point. In any case few references on it should be included, both on biodeterioration problems and potentials of such compound.

 In the introduction a further suggested references is: Price, C. A., & Doehne, E. (2011). Stone conservation: an overview of current research. Getty publications.

Moreover, when describing the characteristic of the 3 tested materials, I suggest the utility of providing a comparative table to enhance the similarity and differences among them (some of them are in the methods, whereas other are in the results) For some of them it is a date and in one case it is missing). More importantly, why such tests were carried out with original materials, and not with samples similar, and more suitable to be used as test?

Few little modifications are suggested:

  • Join the Table 5, 6, 7 in a Single Table
  • Join Fig, 3, 4, 5 in a Sigle Table
  • Use the same sequence when describing the products (caliche, Compañía and Sostenes, or what you prefer...(in the text and in the Tables)
  • In Table 8 the C of Caliche is missing

Author Response

Comments and Suggestions for Authors

Reviewer 1

Comment

The paper analyses the consolidant and hydrophobic formulations treatment using THEOS as greener product with respect the most commonly used Teos. The paper shows original and interesting data. The presentation of data is clear, and the general protocols are sound. Then I suggest the publication, even if some revisions are required.

At first, data on colorimetric changes after the treatments are not discussed and analyses are not done. The assessment of such potential is important, when an application to cultural heritage is welcomed. Therefore, two modification are required: The title: change into “Preliminary evaluation of Glycol......”and into the discussion of such further need should be discussed, including references.

Answers:

It is important to say, and we consider to include in the text, that not apparent colorimetric changes were observed after treatment in any sample, although this was done by simple observation and analyses must be done and considered in perspectives as the use of the Munsell method, commonly applied to observe chromatic variations as a result of the samples treatment. In conclusions (lines 534-540) a new paragraph was incorporated regarding the colorimetric changes after treatments, study that is required to be done as a perspective:

It is very important to point out that data on colorimetric changes after treatments have not been obtained, bearing in mind that the assessment of such analyses is quite important when stones of historical buildings are treated. No apparent colorimetric changes were observed after treatment in any sample, although this was recorded by simple observations. However, corresponding analyses must be performed and considered in perspectives; for example, the use of the Munsell method, which is commonly applied to observe chromatic variations as a result of sample treatment.

As suggested, the title was modified as  “Approach to the use of Glycol Alkoxysilanes-Polysaccharides Hybrids in the Conservation of Historical Building Stones”

Comment

The Author also at first introduce the potential of chitosan in reducing the biological colonization. Then they underline as further study will assess such point. In any case few references on it should be included, both on biodeterioration problems and potentials of such compound.

Answer:

In first term, in the introduction (lines 37-45) we have included more details about stone decay. The complete paragraph was modified as follows:

In an overview of research on stone conservation, the process referred to as consolidation is considered to be an active conservation process “where stone is severely weakened by decay,

In aspects related to antimicrobial activity and biodegradation, information from literature was included (lines 143-163) and 6 new references were incorporated to the list (references 41-46):

…the synergy originating from the alkoxysilane–chitosan interaction in terms of the film formation capacity of chitosan, as well as the antimicrobial activity, suggests the possibility to avoid the biodegradation of stones by many organisms. It is important to consider that the material aptitude to biological colonization by certain organisms is called the bioreceptivity and is dependent on different environmental factors, such as the pH, water availability, climate exposure, mineral composition, porosity, permeability, and nutrient sources [5, 41].

The antimicrobial activity of chitosan has been widely studied; for example, the antibiofilm properties of chitosan-coated surfaces, where chitosan offers a flexible, biocompatible platform for designing coatings to protect surfaces from infection [42]. The state of the art of antimicrobial chitosan and chitosan derivatives and the effects of structural modifications on the activity and toxicity have been reviewed toward improving the understanding of the bioactivity and to develop more useful chitosan conjugates [43]. On the other hand, it has been pointed out that chitosan and its derivatives can be called environmental purification functional materials as they can effectively control the growth and reproduction of hazardous bacteria and also control toxic pollutants [44]. The antibacterial activity of chitosan extracted from a pen shell against both Gram-positive and Gram-negative bacteria was recently reported [45]. Furthermore, chitosan and its derivatives have been studied due to their antimicrobial properties in the context of preventing and treating denture stomatitis, which can be caused by fungi [46].

The following references were incorporated:

41 Warscheid, Th.; Braams, J. Biodeterioration of stone: a review. International Biodeterioration and Biodegradation. 2000, 46, 343-368.

42 Carlson, R.P.; Taffs, R.; Davison, W.M.; Stewart, P.S. Anti-biofilm properties of chitosan-coated surfaces, Journal of Biomaterials Science, Polymer Edition. 2008, 19, 8, 1035-1046. DOI: 10.1163/156856208784909372

43 Sahariah, P.; Másson, M. Antimicrobial Chitosan and Chitosan Derivatives: A Review of the Structure-Activity Relationship. Biomacromolecules. 2017, DOI: 10.1021/acs.biomac.7b01058

44 Atay, H. Y. Antibacterial Activity of Chitosan-Based Systems.  2019, S. Jana, S. Jana (eds.), Functional Chitosan, https://doi.org/10.1007/978-981-15-0263-7_15

45 Sudatta, B.P.; Sugumar, V.; Varma, R. Extraction, characterization and antimicrobial activity of chitosan from pen shell, Pinna bicolor. International Journal of Biological Macromolecules. 2020, https://doi.org/10.1016/ j.ijbiomac.2020.06.291

46 Walczak, K.; Schierz, G.; Basche, S ; Petto, C.; Boening, K.; Wieckiewicz, M. Antifungal and Surface Properties of Chitosan-Salts Modified PMMA Denture Base Material. Molecules. 2020, 25, 5899; doi:10.3390/molecules25245899

The reference of Price, C. A., & Doehne, E. (2011). Stone conservation: an overview of current research. Getty publications, was already in the list as reference [5].

Comment

Moreover, when describing the characteristic of the 3 tested materials, I suggest the utility of providing a comparative table to enhance the similarity and differences among them (some of them are in the methods, whereas other are in the results) For some of them it is a date and in one case it is missing).

Answer:

In Materials and Methods, the mineralogical analysis missing of Compañía was included, as well as a new table (Table 7) to compare the mineralogical composition of the 3 samples as suggested (lines 479-495):

The reported mineralogical analysis indicated alkaline feldspar (65%), quartz (29%), calcium silicate (3%), and hematite (4%) contents [33]. Additionally, a third sample from an archeological site called “Cerro de Los Remedios” (located in Comonfort county, Guanajuato state, México) was studied. As part of the basement of a pyramid, this stone possesses a calcareous nature (caliche), with CaCO3 (93%), CaO (1.5%), halloysite (5%), and traces of hematite. The relatively high amount of kaolinite present in the stone might be taken as evidence that some of the original feldspars have been hydrolyzed to clays by the weathering process known as kaolinization [34]. A comparative composition of the samples is presented in Table 7.

Table 7. Mineralogical composition of the siliceous and calcareous stones.

Mineral

Caliche

Compañía

Sostenes

Alkaline feldspar

-

65%

46%

Quartz

-

29%

27%

Calcite

93%

-

-

CaO

1.5%

-

-

Mica

-

-

10%

Kaolinite

-

-

9%

Calcium silicate

-

3%

-

Halloysite

5%

-

-

Smectite

-

-

3%

Hematite

traces

4%

traces

To evaluate the consolidation effect of the hybrid formulations, some samples (already cleaned and dried) were treated with the THEOS–chitosan formula, and to test the hydrophobic properties, others were treated with the MeTHEOS–chitosan formulation. The formulations were applied on stone samples by brushing in one phase under laboratory conditions until saturation.

Then, all samples were carefully wrapped in black plastic polypropylene sheets (as used in practical conservation) to permit gelling and aging for 2 weeks.

Comment

More importantly, why such tests were carried out with original materials, and not with samples similar, and more suitable to be used as test?

Answer:

We have performed several studies regarding geological and mineralogical analyses of different historical buildings in our area. Because of the complexity and variation in composition, is not easy to find out similar materials or even to prepare them. Our approach has been to go ahead to study directly the real materials with their own (in some cases) particular decay problems.

Comment

Few little modifications are suggested:

Answers:

  • Join the Table 5, 6, 7 in a Single Table
  • Join Fig, 3, 4, 5 in a Sigle Table

Aswers:

            Some tables were corrected as suggested (dynamic contact angle) as well as some figures. Some of them, we consider not to do it because it will result in a quite big figure or table.  

  • Use the same sequence when describing the products (caliche, Compañía and Sostenes, or what you prefer...(in the text and in the Tables)

Answer:

  • The sequence is now the same in text and tables (caliche, Compañía, Sostenes in Tables 1, 2, 3, 4, 5 6, as in the new one 7).  
  • In Table 8 the C of Caliche is missing
  • The correction was done.

Reviewer 2 Report

The paper reports on a potentially interesting topic related to the use of chitosan/alkoxysilanes hybrid mixtures, which could be used for the consolidation and/or protection of stones in the field of cultural heritage. The idea and the premises are interesting, especially with regards to the treatment of siliceous stones. The main advantage is that this would be an aqueous treatment, with a subsequent benefit both for the environment and the operators' health. Secondary advantages were briefly discussed and could be further explored.

However, I'm sorry to say that the manuscript in the present form is not at the same level of previous publications by the same authors (on similar topics), and, thus, I believe that is not suitable for publication. In fact, it shows some weaknesses in its style and presentation (English language should be revised, some sentences are unclear, all the figures are hardly readable), but, most importantly, presents a series of experiments, which are seriously flawed. Or, at least, they are reported (and, possibly, conducted) in a confused way. The meaning and the aim behind some of the measurements is not evident (why performing and reporting EDX analyses on a film of chitosan/alkoxysilane just synthesized by the authors themselves?). Errors and uncertainties related to data, when reported, are ignored, so that the discussion of data has no actual significance (see lines 264-269, on the presence of nitrogen in SEM-EDX measurements). The rationale on which formulations were chosen for the application seems to be completely arbitrary (lines 282-291). No clues were given on the way FTIR-ATR measurements could be performed on solid stone samples (while this technique is usually performed on powders or film-like materials, which can be easily pressed against the crystal where the total reflection of the IR beam is generated)... And so on.

I therefore advise that the paper should be reconsidered only if completely rewritten after having carefully revised (and likely rerun) most of the experiments.

Author Response

Reviewer 2

Comments and Suggestions for Authors

Comments

The paper reports on a potentially interesting topic related to the use of chitosan/alkoxysilanes hybrid mixtures, which could be used for the consolidation and/or protection of stones in the field of cultural heritage. The idea and the premises are interesting, especially with regards to the treatment of siliceous stones. The main advantage is that this would be an aqueous treatment, with a subsequent benefit both for the environment and the operators' health. Secondary advantages were briefly discussed and could be further explored.

However, I'm sorry to say that the manuscript in the present form is not at the same level of previous publications by the same authors (on similar topics), and, thus, I believe that is not suitable for publication. In fact, it shows some weaknesses in its style and presentation (English language should be revised, some sentences are unclear all the figures are hardly readable).

Answers:

In respect to English language, the manuscript was revised by a service given by experts. We expect to full fill the requirement.

On the other hand, all figures were improved in quality (Figures 1, 3, 5, 5 and 6). A new Figure 2 was included for more information regarding FTIR to study the thermal stability of films.

Comments

but, most importantly, presents a series of experiments, which are seriously flawed. Or, at least, they are reported (and, possibly, conducted) in a confused way. The meaning and the aim behind some of the measurements is not evident (why performing and reporting EDX analyses on a film of chitosan/alkoxysilane just synthesized by the authors themselves?).

Errors and uncertainties related to data, when reported, are ignored, so that the discussion of data has no actual significance (see lines 264-269, on the presence of nitrogen in SEM-EDX measurements).

Answers:

Several improvements were done in the manuscript according to the observations and are included or incorporated, or corrected in different paragraphs:

In Results and discussion, we justify why we obtain and characterize the films (lines 166 to 170).

The results and discussion section considers two main aspects: synthesis and characterization of the formulations THEOS–chitosan and MeTHEOS–chitosan and their application in siliceous and calcareous building materials. Films of the silanechitosan hybrids were obtained and characterized using different methods. Such characterization was useful for revealing the film behavior inside or on the stones as a result of the consolidation or hydrophobic treatment.

And also, in section 2.1.2 SEM-EDX (lines 209-214)

2.1.2. SEM–EDX

SEM–EDX of THEOS–chitosan and MeTHEOS–chitosan films are presented in Figure 1. The 10,000× amplification illustrates the films’ characteristics, showing that they are flexible, thin, and transparent and have no evident imperfections. The elements observed according to EDX analysis are carbon, nitrogen, oxygen, and silicon, in accordance with the hybrid composition.

Regarding SEM-EDX we have reviewed the way it was written (lines 295-303) for more clarity:

Regarding EDX analysis, Sostenes and caliche stones display an increment in the carbon and nitrogen atomic percentage following treatment. A plausible interpretation is that the chitosan chains are exposed to the surface, not just in the case of the consolidant, but in the hydrophobic treatment (the methyl groups are surface oriented). Additionally, it is interesting to observe that the nitrogen atomic percentage is higher in consolidated Sostenes stone than in caliche, suggesting that the interaction between the consolidant and caliche possibly occurs via the free amine group. On the other hand, in Compañía stone, silicon is the element with a major atomic concentration on surface, probably suggesting, in accordance with SEM, that not enough consolidant formulation was added.

The section 2.1.1 (Hardness determination, lines 304-320) includes some corrections in respect to the Mohs scale, and also, to specify the formulations used (in correlation to Materials and Methods where formulations 1, 2 and 3 were described).

2.2.1. Hardness determination

The effectiveness of treatment in terms of the mechanical properties was determined by hardness measurements in stones consolidated using the THEOS–chitosan formulation and was performed by indentation with a Shore D durometer. Three variables that influence the hardness increase were considered: the applied formulation (as a function of the silane/chitosan ratio); the nature of the stone; and the percentage of deacetylation of the chitosan used in the formulation (%DDA). A statistical analysis was conducted to evaluate the effect of each variable (not included here). In a next step, the Shore D hardness values were transformed to the most common hardness scale, such as Vickers, Brinell, and finally Mohs, in order to compare the hardness data obtained with respect to reference values of well-studied materials based on the Mohs scale.

The formulations named 1, 2, and 3 (see Materials and Methods) were used in hardness determination. The hardness was measured at four points of the samples before and after treatment to characterize the hardness percentage increase. Interesting results were obtained for every formulation; however, the treatment that remarkably increased the mechanical properties of the stones was formulation 2, which contains chitosan with 66% DDA (Table 1).

The paragraph (lines 331-337) was also reviewed in some way as a general conclusion of hardness determination. 

The hardness values transformed to Mohs scale and reported in table 2, indicate a hardness increment of 1 unit in siliceous materials and in the case of caliche (formulations 1 and 2) even 2 units. In general, the most important increase in hardness is in caliche. In terms of the Mohs scale, the hardness values from 5 to 7 obtained for the samples, range between apatite to orthoclase and quartz. The hardness studies indicate that the samples treated with THEOS-chitosan leads to an important increase in the mechanical properties of the three materials.

Some clarification about the use of formulations 1, 2 and 3 in the evaluation of the formulation MeTHEOS-chitosan by contact angle measurements was also done (lines 352-357).

2.2.3. Contact angle measurements.

The evaluation of the hydrophobic formulation MeTHEOS–chitosan was studied by static and dynamic contact angle measurements using the same formulations 1, 2, and 3 and the % of DDA of 66. Because of the natural existence of defects on certain materials, as is the case of the stones studied in the current investigation, it has been suggested that a static water contact angle does not necessarily characterize the intrinsic water wettability [40]. Dynamic contact angle determination in the three stones is presented. The dynamic contact angle was obtained by the degree of hydrophobicity calculated by the hysteresis, representing the difference as θR (receding angle) − θA (advancing angle). The hysteresis values and the average of three measurements in different surface sections of the three stones are reported in Table 4. The dynamic angle measurements indicate that the surfaces of the three stones studied display water repellency.

Comment

The rationale on which formulations were chosen for the application seems to be completely arbitrary (lines 282-291).

Answer:

We consider very important to explain why and how the formulations were selected for the different experiments performed in the application and no in an arbitrary way: Two sections were corrected, in Results and Discussion (lines 185-291):

2.1. Synthesis and Characterization of silane–chitosan hybrid films

A very wide range of tests of THEOSchitosan and MeTHEOSchitosan solutions were prepared, using different proportions of the reagents, in order to find out the most appropriate formulations to be applied in stone treatment. The selection of formulations to be used was carried out through observation of the film characteristics obtained in terms of the flexibility, homogeneity, transparency, and resistance to syneresis, where the excellent capability of chitosan as a film formulation was a key aspect in terms of the concentration of chitosan used. No phase separation was observed. The extensive testing revealed that a selection of formulations with similar physical characteristics were obtained, offering the possibility to apply them in experiments with different goals (see Materials and Methods). For example, the selected formulation for the films characterized by FTIR, scanning electron microscopy (SEM)EDX, thermal stability, and solid state NMR analysis was based on 10 mL of an aqueous solution of chitosan (0.5% in acetic acid at 1% with 72% deacetylation) and 0.5 g of THEOS. Some variations in the formulation composition were used in several analyses, such as hardness and contact angle determinations, in order to study the effect of the silane–chitosan ratio (formulations referred to as 1, 2, and 3 in Materials and Methods).

2.1. Synthesis and Characterization of hybrid films silane-chitosan

2.1. Synthesis and Characterization of silane–chitosan hybrid films

A very wide range of tests of THEOSchitosan and MeTHEOSchitosan solutions were prepared, using different proportions of the reagents, in order to find out the most appropriate formulations to be applied in stone treatment. The selection of formulations to be used was carried out through observation of the film characteristics obtained in terms of the flexibility, homogeneity, transparency, and resistance to syneresis, where the excellent capability of chitosan as a film formulation was a key aspect in terms of the concentration of chitosan used. No phase separation was observed. The extensive testing revealed that a selection of formulations with similar physical characteristics were obtained, offering the possibility to apply them in experiments with different goals (see Materials and Methods). For example, the selected formulation for the films characterized by FTIR, scanning electron microscopy (SEM)EDX, thermal stability, and solid state NMR analysis was based on 10 mL of an aqueous solution of chitosan (0.5% in acetic acid at 1% with 72% deacetylation) and 0.5 g of THEOS. Some variations in the formulation composition were used in several analyses, such as hardness and contact angle determinations, in order to study the effect of the silane–chitosan ratio (formulations referred to as 1, 2, and 3 in Materials and Methods).

And in section 3.2.2 of Materials and Methods (lines 421-434)

3.2.2. Synthesis of THEOS–chitosan and MeTHEOS–chitosan solutions

Formulation solutions were prepared by the addition of 0.5 g of THEOS or MeTHEOS to 10 mL of an aqueous solution of chitosan (0.5%) in acetic acid (1%) under magnetic stirring until complete dissolution. The % of DDA of chitosan was 72% (Sigma Aldrich). The described solutions were used to obtain hybrid films that were characterized by FTIR, SEMEDX, and solid state NMR and regarding their thermal stability, and then used in consolidation and hydrophobic treatments of the stones. The formulations named 1, 2, and 3 were prepared using chitosan obtained from the extraction of shrimp exoskeleton with three different degrees of deacetylation (%DDA): 62%, 66%, and 70%. Formulation 1 (0.5 g of THEOS or MeTHEOS and 10 mL of 0.5% aqueous solution of chitosan), formulation 2 (1 g of THEOS or MeTHEOS and 10 mL of a 0.5% aqueous solution of chitosan), and formulation 3 (0.5 g of THEOS or MeTHEOS and 10 mL of a 1% aqueous solution of chitosan) were applied to the stones and used in the hardness and contact angle determinations.

Comments

No clues were given on the way FTIR-ATR measurements could be performed on solid stone samples (while this technique is usually performed on powders or film-like materials, which can be easily pressed against the crystal where the total reflection of the IR beam is generated)... And so on.

Answer:

In Materials and Methods (section 3.3.2) the information about FTIR-ATR measurements (lines 457-462) has been included:

3.3.2. FTIR analysis

The spectra of hybrid films from 4000 to 650 cm−1were collected using a Perkin Elmer Spectrum FTIR 1600 coupled with an ATR accessory (germanium point, 100 μm in diameter). An average of 16 scans was obtained, with a resolution of 4 cm−1. Similar experimental conditions were used in the case of treated and untreated stone samples. The FTIR spectra were obtained from powders (−100 mesh) of each stone.

Comment

I therefore advise that the paper should be reconsidered only if completely rewritten after having carefully revised (and likely rerun) most of the experiments.

Answer:

We expect the general written revision, the corrections and modifications done, and the explanation of them can full fill the reviewer comments and observations.

Reviewer 3 Report

This is an interesting manuscript on the use of Glycol Alkoxysilanes-Polysaccharides on building stone as consolidators. The authors show a good knowledge of the subject and they have a good track. The manuscript s well written and provides interesting information. My only concern refers to figures. The figures provided are insufficient and too small. Figures 1, 3, 4 and 5 must be amplified. They cannot be seen in the present format. Moreover, more figures of FTIR are required. The authors refer to FTIR results in section 2.1.1., 2.1.3, 2.2., … but they just provide a figure for section 2.2.

After such minor changes the manuscript will be ready for publication.

Author Response

Reviewer 3

Comments

This is an interesting manuscript on the use of Glycol Alkoxysilanes-Polysaccharides on building stone as consolidators. The authors show a good knowledge of the subject and they have a good track. The manuscript s well written and provides interesting information. My only concern refers to figures. The figures provided are insufficient and too small. Figures 1, 3, 4 and 5 must be amplified. They cannot be seen in the present format. Moreover, more figures of FTIR are required. The authors refer to FTIR results in section 2.1.1., 2.1.3, 2.2., … but they just provide a figure for section 2.2.

After such minor changes the manuscript will be ready for publication.

Answers:

Figures 1, 3, 4 and 5 were amplified as suggested.

2.1.1. The FTIR of the films have been included in Supplementary information. Previously it was the Figure S1. FTIR-ATR for films of (a) MeTHEOS-Chitosan and (b) THEOS-Chitosan. 2.2. We have incorporated Figure S6. FTIR-ATR spectrum of Compañia sample without treatment, consolidated and hydrofugated and Figure S7. FTIR-ATR spectrum of Sostenes sample without treatment, consolidated and hydrofugated

2.1.2. SEM-EDX. The figures have been improved

2.1.3. A representative FTIR has been incorporated of FTIR of MeTHEOS-chitosan at 25 and 350 °C, in order to compare the differences in respect to thermal stability. Figure 2. FTIR-ATR for a film of MeTHEOS-Chitosan at a) 25 °C and b) 350 °C

A new sentence was written (lines 219-224):

2.1.3. Thermal stability of the hybrids

The thermal stability of the hybrid films was studied. The films were exposed to different temperatures, from room temperature to 700 °C, and FTIR–ATR spectra were collected to determine any structural changes. Comparative spectra obtained at 25 and 350 °C are presented (Figure 2). As can be observed, the films are thermally stable until 350 °C. In the case of MeTHEOS–chitosan, the fragment –SiCH3 is removed around 500 °C.

Round 2

Reviewer 2 Report

The manuscript was certainly improved after its revision by the authors. However, the figures still need to be edited in order to make them well readable. Writings and numbers (if necessary) must be rewritten using a suited font size, while, when not strictly necessary, should be eliminated.

My advice is to publish the paper in the present form after this very minor, yet significant, graphical editing.

Author Response

Reviewer 2

Comments:

The manuscript was certainly improved after its revision by the authors. However, the figures still need to be edited in order to make them well readable. Writings and numbers (if necessary) must be rewritten using a suited font size, while, when not strictly necessary, should be eliminated.

My advice is to publish the paper in the present form after this very minor, yet significant, graphical editing.

Answers:

In accordance to reviewer 2 suggestions, the figures have been improved in size, and we consider they are in a much more better view in order to make them well readable.

The font in figures labels has been changed to a better size.

We wish to thank reviewers for their comments and suggestions to improve the manuscript.